# Girgentana’s Goat Milk Microbiota Investigated in an Organic Farm During Dry Season

**DOI:** 10.3390/ani15213149

**Published:** 2025-10-30

**Authors:** Giorgio Chessari, Serena Tumino, Bianca Castiglioni, Filippo Biscarini, Salvatore Bordonaro, Marcella Avondo, Donata Marletta, Paola Cremonesi

**Affiliations:** 1Dipartimento di Agricoltura, Alimentazione e Ambiente, Università di Catania, 95123 Catania, Italy; giorgio.chessari@unict.it (G.C.); serena.tumino@unict.it (S.T.); s.bordonaro@unict.it (S.B.); mavondo@unict.it (M.A.); donata.marletta@unict.it (D.M.); 2Istituto di Biologia e Biotecnologia Agraria, National Research Council, 26900 Lodi, Italy; filippo.biscarini@cnr.it (F.B.); paola.cremonesi@cnr.it (P.C.)

**Keywords:** goat milk, milk microbiota, 16S rRNA, local goat breed

## Abstract

**Simple Summary:**

Milk microbiota is a complex microbial ecosystem harboring a diverse community of microorganisms which can influence milk quality, safety, and flavor, and may also play a role in animal health and udder physiology. The Girgentana goat is an ancient dairy breed native to Sicily, known for its good production of high-quality milk intended for direct consumption or processing into ricotta or cheese. Girgentana’s milk microbiota has been investigated for the first time in an organic farm located in Agrigento area. Illumina NovaSeq technology was used to sequence the V3–V4 regions of 16S rRNA from 44 individual and 3 bulk milk samples. Alpha diversity metrics were estimated to assess richness, evenness, and overall milk microbiota diversity, while taxa composition was analyzed to characterize microbial structure. This study is the first to unravel the milk microbiota composition of the Girgentana breed, providing detailed microbial profiling. Moreover, this can represent a baseline for microbial diversity, which can be used for future comparisons with other breeds raised in traditional dairy systems.

**Abstract:**

Milk microbiota is a complex microbial ecosystem with implications for product quality, safety, and animal health. However, limited data exist on goat milk microbiota, particularly in local breeds. This study provides the first detailed characterization of the milk microbiota of Girgentana goats, a resilient Sicilian breed valued for high-quality dairy products. Illumina NovaSeq sequencing was used to analyze the 16S rRNA V3–V4 regions of 44 individual and 3 bulk milk samples. Briefly, 16S rRNA-gene sequencing produced a total of 8,135,944 high-quality reads, identifying 1134 operational taxonomic units (OTUs) across all individual samples. On average, each sample showed 864 OTUs with counts > 0. Alpha diversity metrics, based on richness estimators (Chao1: 948.1; ACE: 936.3) and diversity indices (Shannon: 4.06; Simpson: 0.95; Fisher: 118.5), indicated a heterogeneous community with both common and low-abundance taxa. *Firmicutes* (51%) and *Proteobacteria* (27%) were the predominant phyla, with *Lactobacillaceae* (54%) and *Bifidobacteriaceae* (22%) dominating at the family level. Notably, farm bulk milk profiles closely mirrored individual samples. These results establish a milk microbiota baseline for the Girgentana breed and offer valuable insights into microbial ecology in traditional dairy systems, supporting future comparisons across breeds and farming practices.

## 1. Introduction

In recent years, the study of milk microbiota in dairy farming has gained significant attention, not only for its influence on the organoleptic properties of raw milk but also for its crucial role in dairy processing technologies [1]. Advances in analytical methods, combined with the emergence of omics-based approaches, have reshaped our understanding of milk as a dynamic and diverse microbial ecosystem with multiple biological functions [1,2]. There is growing recognition of the association of milk microbiota with animal health, especially as it responds to environmental changes [3,4] or shifts in physiological states [5,6]. As such, it is increasingly regarded as a valuable omics-based indicator of host well-being. However, the composition of milk microbiota is rather heterogeneous and varies considerably across species, influenced by a range of factors, including diet [2], management practices [7], breed [5], and overall health status [2,5]. Although goat milk production represents a key component of dairy systems worldwide, studies on its microbial composition remain limited. This gap is even more pronounced when considering local and indigenous goat breeds, which play a vital role in supporting rural economies and promoting sustainable farming practices in diverse geographic regions [8,9,10,11]. Across Europe and globally, native goat breeds represent a unique reservoir of genetic diversity, particularly valuable in marginal environments where they contribute to biodiversity preservation, ecosystem services, and traditional dairy production [12].

Girgentana is a Sicilian ancient breed which holds notable significance in its regional context, contributing to the cultural heritage, serving as a valuable genetic resource [13,14]. It has been widely recognized for its high-quality milk, adaptability to arid and harsh environments, and disease resistance. Indeed, previous studies on Girgentana goats have focused primarily on genetic and productive traits [13,15], highlighting their outstanding adaptability and resilience. Typically raised in semi-extensive farming systems within harsh and marginal areas, the Girgentana goat is often subject to limited monitoring, which results in scarce data on the environmental factors (such as diet, climate conditions, and pathogen exposure), that may impact both animal health and milk composition. However, no study so far has explored their milk microbiota, despite its potential to reveal functional aspects related to milk quality, health, and microbial terroir.

Understanding the milk microbiota of indigenous breeds such as the Girgentana goat is important to evaluate how microbial community structure may influence milk quality and safety, impacting the cheese-making processes and sensory characteristics of traditional dairy products [16], as well as reflecting animal health and udder status, offering a non-invasive marker of mammary gland conditions [17]. Finally, microbial signatures can serve as traceability tools, helping to differentiate products from specific breeds or farming systems and to protect local designations of origin [18].

Recent studies have strengthened the link between production systems and milk microbiota composition, reporting significant farm-to-farm variability in raw goat milk microbiota between conventional and traditional systems in The Netherlands [19]. Similarly, Montesinos Rivera et al. [20] demonstrated that milking stage and management type (semi-intensive vs. traditional) significantly affected the alpha and beta diversity of goat milk microbiota in Mexico. Furthermore, recent work by Zhang et al. [21] highlighted that health status, particularly mastitis, can markedly alter goat milk’s microbiota structure, providing essential microbiological insights that could inform effective prevention and treatment strategies for mastitis in dairy goats.

Culture-independent molecular techniques, particularly those based on sequencing of the 16S ribosomal RNA (rRNA) gene, are commonly employed to investigate the bacterial diversity present in milk [6]. These advanced approaches enable a more comprehensive characterization of microbial communities, uncovering their complexity and dynamic nature, which often remain undetected by traditional culture-based methods. A range of bioinformatics tools and software platforms, including QIIME [22], MICCA [23], and DADA2 [24], are routinely used to process 16S rRNA gene sequencing data. These pipelines facilitate key analytical steps such as data filtering, sequence alignment, taxonomic classification, and statistical analysis, ultimately providing in-depth insights into the diversity, structure, and potential function of milk microbiota.

The present study provides the first characterization of the milk microbiota of the Girgentana goat breed reared in an organic farm under semi-extensive organic conditions during the dry season. Using high-throughput sequencing of the 16S rRNA gene (V3–V4 regions), we aimed to describe microbial diversity and composition, identify dominant and low-abundance taxa, and establish a baseline profile for this local breed.

## 2. Materials and Methods

### 2.1. Animals and Sample Collection

The study was conducted on an organic, semi-extensive goat farm located in the Agrigento area, Sicily, Italy. Samples were collected from a larger herd of approximately 250–280 Girgentana goats. A total of 53 lactating goats were selected based on homogeneous lactation stage (120 ± 5 days in milk), similar milk yield (889.6 ± 188.1 g/day), and apparent good health, with no clinical signs of mastitis. Somatic cell counts (SCC) averaged 690 × 10^3^ cells/mL. While SCC thresholds are well defined in cows, no official reference limits exist for goats, and higher values are common at late lactation [25].

Individual milk samples were collected at the end of May, together with three bulk milk samples obtained from the same herd. At the time of sampling (late May 2022), local climate conditions were recorded: mean temperature 25.1 °C, minimum 20.2 °C, maximum 31.6 °C, and relative humidity 26%. For comparison, the monthly averages for May 2022 in this area were 18.2 °C (mean), 13.6 °C (minimum), 22.5 °C (maximum), and 63.5% relative humidity. In accordance with the season and the semi-extensive management system, the animals had daily access to pasture (6–8 h/day) and were fed a diet primarily composed of organic mixed hay offered ad libitum and 700 g of a supplement based on 50% of organic barley whole grain and 50% of faba bean whole grain offered twice a day during milkings (in the hay and in the supplement: crude protein, 12.5% and 18.7% DM: NDF, 55.0% and 22.2% DM).

Goats were milked twice daily, and all samples for this study were collected during a single morning milking session. To minimize contamination, the first streams of foremilk were manually discarded, and the teat ends were thoroughly cleaned. Approximately 10 mL of milk was then aseptically collected from each animal. All samples were transported to the laboratory at 4 °C, and stored at −20 °C until DNA extraction [26].

### 2.2. DNA Extraction, Library Preparation and Sequencing

Analyses for DNA extraction (starting from 2 mL of milk sample) and library preparation were performed at the laboratory of molecular biology of Istituto di Biologia e Biotecnologia Agraria (IBBA) in Lodi, following the protocols proposed by Cremonesi et al. [27]. DNA was extracted by using a method based on the combination of a chaotropic agent, guanidium thiocyanate, with silica particles, to obtain bacterial cell lysis and nuclease inactivation. DNA quality was assessed using a NanoDrop ND-1000 spectrophotometer (NanoDrop Technologies, Wilmington, DE, USA). Bacterial DNA was amplified using the specific amplicon-primers, which target the V3–V4 hypervariable regions of the 16S rRNA gene: 16S Amplicon PCR Forward Primer = TCGTCGGCAGCGTCAGATGTGTATAA-GAGACAGCCTACGGGNGGCWGCAG; 16S Amplicon PCR Reverse Primer = GTCTCGTGGGCTCGGAGATGTGTATAAGAGACAGGACTACHVGGG-TATCTAATCC). All PCR amplifications were performed in 25 μL volumes per sample. A total of 12.5 μL of Phusion High-Fidelity Master Mix 2× (Thermo-Fisher Scientific, Walthem, MA, USA) and 0.2 μL of each primer (100 μM) were added to 2 μL of genomic DNA (5 ng/μL). In addition to blank controls (no DNA template added to the reaction), positive internal lab controls consisting of DNA extracted from milk microbiota samples were also included to verify amplification performance.

The amplification step was performed in an Applied Biosystem 2700 thermal cycler (Thermo-Fisher Scientific). Samples were denatured at 98 °C for 30 s, followed by 25 cycles with a denaturing step at 98 °C for 30 s, annealing at 56 °C for 1 min and extension at 72 °C for 1 min, with a final extension at 72 °C for 7 min. Amplicons were cleaned with Agencourt AMPure XP (Beckman, Coulter Brea, CA, USA) and libraries were prepared following the 16S Metagenomic Sequencing Library Preparation Protocol (Illumina, San Diego, CA, USA). DNA concentration and fragment size were measured on a Qubit fluorometer (Invitrogen, Carlsbad, CA, USA) and Agilent Bioanalyzer (Santa Clara, CA, USA), respectively. Sequencing was performed using NovaSeq (Illumina) technology with 2×300-base paired-end reads, outsourced to a specialized service provider (Nuova Genetica Italiana S.r.l., Como, Italy).

### 2.3. Bioinformatic Pipelines and Quality Control

A total of 7 out of 53 individual milk samples failed to undergo sequencing due to issues encountered during the amplification process. Additionally, two samples yielded an insufficient number of reads during the sequencing step. Consequently, 44 individual samples, along with 3 bulk milk aliquots, were retained for downstream analysis.

Demultiplexed paired-end reads from 16S rRNA-gene sequencing were first checked for quality using MultiQC v1.17 [28] for an initial assessment. Adapters and primers were removed with Cutadapt v4.4 [29]. Reads from 16S rRNA-gene sequencing were processed with MICCA v1.7 [23] in-house pipelines. Before joining forward and reverse paired-end reads, sequences were trimmed for low-quality base calls (Phred < 25) using the software package sickle v1.33 [30]. MICCA pipeline was used to join the reads and to filter for unknown bases. All remaining reads were combined in a single FASTA file and processed with MICCA v1.7 using the de novo UNOISE method (*denovo_unoise*). This approach denoises sequences, bins them based on identity without reference to a database, removes chimeras, and generates high-quality operational taxonomic units (OTUs) for identification and quantification of the milk microbiota.

Taxonomy assignments were performed using the RDP classifier v2.12 with a Bayesian approach and the RDP 18 reference database, in order to estimate the probability of a 16S rRNA sequence belonging to a particular taxonomic group [31]. A pre-defined taxonomy map of reference sequences to taxonomies was then used for taxonomic identification along the main taxa ranks down to the genus level (domain, phylum, class, order, family, genus). By counting the abundance of each OTU, the OTU table was created and then grouped at each phylogenetic level. OTUs with total counts lower than 15 in fewer than 2 samples were filtered out.

### 2.4. Microbiota Characterization and Composition

Microbial diversity was assessed both within (alpha diversity) and between (beta diversity) samples. All diversity indices were calculated from the complete OTU table at the OTU level using the *phyloseq* package v1.38.0 [32] in R v4.1.2 environment [33], applying default parameters. Six alpha diversity indices were computed for both individual and bulk milk samples. To evaluate species richness, the Observed, Chao1, and ACE indices were used [34,35,36]. Community evenness was assessed using the Fisher index [37], while the Shannon and Simpson indices [38] were applied to capture both richness and evenness components.

To account for differences in sequencing depth, OTU counts were normalized for beta diversity analyses using Cumulative Sum Scaling (CSS) [38]. Beta diversity was then assessed exclusively on the individual milk samples (*n* = 44), using Bray–Curtis [39] and Euclidean dissimilarity matrices. Ordination analyses (PCoA) were conducted to explore overall community structure and visualize compositional patterns among samples.

## 3. Results

### 3.1. Sequencing Metrics and Quality Filtering

Sequencing of the final dataset, consisting of 44 individual and three bulk milk samples, produced a total of 16,268,414 paired-end reads, with an average read length of 250.60 bp and a mean GC content exceeding 52%. Following quality filtering, a total of 8,672,881 high-quality reads were retained, corresponding to a reduction of 46.7% compared to the raw data, with an average number of paired-reads of 184,529.4 per sample. A detailed summary of paired-read counts across all processing steps is provided in Table 1.

### 3.2. OTUs and Taxa Abundancy

After filtering out OTUs with fewer than 15 counts in at least two samples, the final number of OTUs was 1134 for individual milk samples and 352 for bulk milk samples. For the 44 individual samples, the number of observed OTUs was plotted as a function of both sequencing depth (number of reads up to 250,000) and sample size, resulting in sequence-based and sample-based rarefaction curves (Appendix A). Both curves asymptotically approached a plateau, indicating that the sequencing depth and the number of samples were sufficient to capture the microbial diversity of milk in this study. Further sequencing or additional samples would likely provide only a minimal increase in the number of newly detected OTUs.

Taxonomic classification of OTUs from phylum to genus level revealed differences in microbial diversity between individual and bulk milk samples. In individual milk samples, a total of 7 phyla, 11 classes, 20 orders, 29 families, and 54 genera were identified. In contrast, bulk milk samples showed a lower taxonomic richness, with 5 phyla, 8 classes, 11 orders, 11 families, and 24 genera.

The seven most abundant taxa at each taxonomic level are reported in Appendix A. In individual milk samples, the *Firmicutes* phylum accounted for approximately half of the microbial community (51.27%), followed by *Proteobacteria* (27.37%) and *Actinobacteria* (20.80%), together representing over 45% of the total composition (Figure 1A). Bulk milk showed a similar phylum-level composition (Figure 1B), with *Firmicutes* again being dominant (51.49%), and *Proteobacteria* (24.21%) and *Actinobacteria* (24.14%) present in nearly equal proportions. At deeper taxonomic levels, two families were consistently prevalent in both sample types: *Lactobacillaceae*, with a relative abundance exceeding 50%, and *Bifidobacteriaceae*, accounting for more than 20%. Among genera, *Lactobacillus* (>55%), *Bifidobacterium* (>22%), and *Gilliamella* (>9%) were the most representative across both individual and bulk milk samples (Figure 1C and Appendix A).

### 3.3. Diversity Indices

Alpha diversity metrics were computed to evaluate microbial richness and evenness within milk samples (Table 2). On average, individual milk samples showed higher species richness compared to bulk milk samples, as indicated by the Observed (864.4 vs. 324.0), Chao1 (948.1 vs. 353.9), and ACE (936.3 vs. 342.4) indices. Similarly, the Fisher index, which reflects species diversity, was greater in individual samples (118.5) than in bulk samples (38.8). Regarding community evenness, the Shannon index was slightly higher in individual samples (4.06) compared to bulk samples (3.97), while the Simpson index was comparable between the two sample types (0.95 vs. 0.96). Alpha diversity indices across individual milk samples (*n* = 44) are also reported in Figure 2.

Beta diversity was assessed using Principal Coordinates Analysis (PCoA) based on both Bray–Curtis and Euclidean distance metrics to capture different aspects of microbial community differences. The Bray–Curtis-based PCoA explained 51.0% and 15.7% of the variance on the first and second axes, respectively, while the Euclidean-based PCoA explained a higher proportion of variance, with 65.4% and 15.2% on the first two axes (Appendix A). PCoA plots based on both Bray–Curtis and Euclidean dissimilarity matrices showed no clear clustering patterns, indicating high inter-individual variability in microbial community composition. The three bulk sampling replicates occupy an intermediate position among the individual samples, as expected, since the bulk samples represent composite microbial communities from multiple individuals.

## 4. Discussion

Omics technologies are increasingly advancing in culture-independent techniques, providing increasingly detailed insights into the characterization of microbial communities within a given environment [2,5]. In this study, a sensitivity analysis of the sequence quality filtering pipeline (Phred > 25) was applied to reduce sequencing artifacts, demonstrating the overall robustness of the results obtained through a de novo denoising approach (UNOISE). The observed number of OTUs was plotted as a function of sequencing depth and sample size (Appendix A). Both rarefaction curves approached a plateau asymptotically, suggesting that the sequencing effort and number of samples were sufficient to capture most of the microbial diversity in Girgentana goat milk [40]. This provides a solid foundation for the first in-depth taxonomic profiling of the milk microbiota in this ancient Sicilian breed.

Taxonomic analysis revealed a structured and diverse microbial composition in individual milk samples, with 7 phyla and 54 genera identified, compared to 5 phyla and 24 genera in bulk milk. At the phylum level, both sample types were dominated by *Firmicutes* (around 51%), followed by *Proteobacteria* and *Actinobacteria* in varying proportions. As observed in other ruminant species, *Firmicutes* dominated the microbial community, followed by *Proteobacteria* and *Actinobacteria* [3,6,9,17]. However, the relative abundance of *Bacteroidetes* was negligible, which could be attributed to interspecies microbial competition or specific environmental and dietary factors typical of the semi-extensive farming system adopted here [4,11].

A recent comprehensive review on livestock milk microbiota reported comparable taxonomic patterns, with *Firmicutes* as the most dominant phylum (>50%), followed by *Proteobacteria* and *Actinobacteria* [26], a distribution also typical of goat milk during the milking stage [20]. Although the main phyla identified in Girgentana milk were consistent with those observed in other local goat [19] and sheep breeds [41], the proportional distribution of genera varied. Such variation likely reflects the influence of environmental and management-related factors, including geographical location, feeding regime, and animal health status [6]. Notably, our microbial profile partly overlapped with that of healthy goats described by Zhang et al. [21], who reported a marked reduction in beneficial genera such as *Lactobacillus* and *Bifidobacterium* in mastitic milk of goat individuals. The abundance of these genera in Girgentana milk therefore suggests a microbiota composition compatible with good udder health.

Interestingly, *Actinobacteria* were particularly abundant in both individual and bulk milk. Within this phylum, taxa such as *Bifidobacterium* are technologically and nutritionally valuable, contributing to probiotic potential and beneficial fermentative properties. However, other *Actinobacteria* have been associated with sensory defects during milk storage, including the development of unpleasant bitter or moldy flavors after only a few hours [42]. Under certain conditions, their metabolic activity may also alter lactose content, potentially affecting cheese yield and texture [8]. The relatively high abundance of *Actinobacteria* in Girgentana milk therefore underscores the importance of timely milk processing and highlights possible trade-offs between beneficial and spoilage-associated taxa in raw milk.

Worth of note is the detection of genera such as *Gilliamella*, *Snodgrassella*, and *Frischella*, typically found in the gut of bees and other insects [43,44]. Their presence raises the hypothesis of environmental or vector-borne contamination, likely due to insect activity during milking or handling. While these genera may not pose a direct health risk, their detection highlights the need for stricter hygiene protocols, especially in open-air systems or traditional contexts. Conversely, the high abundance of *Lactobacillus* and *Bifidobacterium* (>55% and >22%, respectively) underlines the potential nutritional and functional value of Girgentana milk, supporting its use in artisanal dairy products and potentially offering probiotic properties [45,46]. These genera are known for their roles in dairy processing, flavor development, and inhibition of spoilage or pathogenic bacteria through acidification and bacteriocin production [47]. Moreover, both *Lactobacillus* and *Bifidobacterium* species have been associated with health-promoting properties, including improved gut barrier function and modulation of host immunity [48,49]. Their high abundance may therefore contribute not only to the technological suitability of Girgentana milk for traditional dairy products but also to its potential nutritional and probiotic value.

In microbiome studies, alpha diversity reflects the richness and evenness of microbial species within a single sample or environment [4,5,40]. Alpha diversity analysis showed that individual samples displayed significantly higher richness (i.e., Observed species, Chao1, and ACE index) and diversity (i.e., Fisher, Shannon and Simpson index) than bulk milk, as indicated by all computed indices. This observation reinforces the importance of individual-level sampling to uncover low-abundance or animal-specific taxa that could be masked in pooled samples. Such taxa may have functional roles, for example in pathogen inhibition, immune modulation, or dairy fermentation, and their loss in bulk analysis may limit interpretability in precision livestock or dairy applications. The slightly lower alpha diversity in bulk samples likely results from a dilution effect, where microbial taxa that are rare or specific to individual animals are either underrepresented or lost when milk is pooled. This supports the idea that relying solely on bulk sampling may underestimate the true microbial complexity of the herd, especially regarding low-abundance taxa that may have functional or health-related significance. For instance, these findings are consistent with previous studies in dairy goats, where healthy milk samples showed higher richness and diversity compared to samples affected by mastitis [21]. In semi-intensive systems, the fore-stripping stage of Mexican goats showed relatively low bacterial diversity (Shannon = 1.5; Simpson = 0.5), whereas the milking stage exhibited higher values (Shannon = 4.0; Simpson = 0.8), reflecting an increase in community complexity during milk ejection [20]. The Shannon (4.06) and Simpson (0.95) indices observed in Girgentana milk are therefore indicative of a highly diverse and evenly distributed microbial community, consistent with healthy animals and efficient hygienic milking practices.

Regarding beta diversity, which assesses differences in microbial composition between samples, PCoA based on Bray–Curtis and Euclidean distances revealed no distinct clustering, suggesting high inter-individual variability, consistent with previous studies in goats and cows [4,5]. As expected, bulk milk samples occupied an intermediate position in ordination plots, reflecting their composite nature. However, it is worth of notice that the limited number of bulk replicates (*n* = 3) may have restricted the ability to draw more robust comparisons. Moreover, the Bray–Curtis dissimilarity metric, which is robust for microbiome data due to its emphasis on relative abundance, is more appropriate than Euclidean distance, which can be overly sensitive to outliers unless data are transformed (e.g., log, Hellinger) [50]. Nonetheless, we included Euclidean-based PCoA to explore potential patterns and observed that it explained a greater portion of variance, although no clear grouping emerged. In summary, these analyses underscore that individual-level sampling not only reveals greater microbial richness but also captures the natural variability in community structure.

From a broader perspective, this study provides a valuable baseline for future comparisons with other goat breeds or dairy systems. In fact, Girgentana goats are managed under low-input, semi-extensive conditions that may promote the establishment of specific microbial communities. Despite providing the first in-depth characterization of the Girgentana goat milk microbiota, the single season (late May, dry season) and a single morning milking may not capture temporal fluctuations in microbial communities. Moreover, environmental sources of contamination, such as insects or feed, may have influenced microbial composition, as suggested by the detection of *Gilliamella* and *Snodgrassella*.

Previous work has shown that environmental temperature, feed availability, and physiological changes across lactation can significantly impact microbial profiles [3,7,11]. Therefore, future research should focus on longitudinal studies, encompassing multiple seasons and lactation stages, to evaluate the stability and resilience of milk microbiota under varying environmental and physiological conditions [51]. Functional metagenomics or metatranscriptomics could also provide insights into the metabolic potential and functional roles of microbial taxa, including lactic acid bacteria, in milk quality, health, and dairy fermentation processes [20,52]. Comparative studies across different breeds, production systems, and feeding strategies would further clarify how management and diet shape milk microbial communities, supporting breed conservation and milk valorization strategies. Such comparisons could also include the same breed under different management systems, to better evaluate the effects of diet, housing, and environmental exposure on milk microbiota.

## 5. Conclusions

This first characterization of the Girgentana goat milk microbiota revealed a diverse microbial community dominated by lactic acid bacteria. The observed microbial profile likely reflects adaptation to local semi-extensive farming conditions and may contribute to the nutritional and functional properties of milk. Future studies integrating functional metagenomics, longitudinal sampling across seasons, and comparisons with other breeds will be essential to elucidate the role of these microbial ecosystems in product quality, animal health, and breed resilience.

## Figures and Tables

**Figure 1 animals-15-03149-f001:**
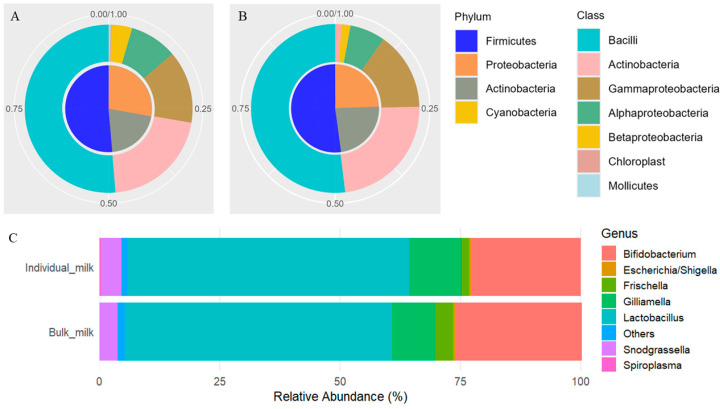
Microbial composition of individual (**A**) and bulk (**B**) goat milk samples at the phylum (inner ring) and class (outer ring) taxonomic levels. Each pie chart displays the relative abundances of the most prevalent taxa. (**C**) Stacked bar plots of the relative abundances of the dominant bacterial genera in individual and bulk milk samples.

**Figure 2 animals-15-03149-f002:**
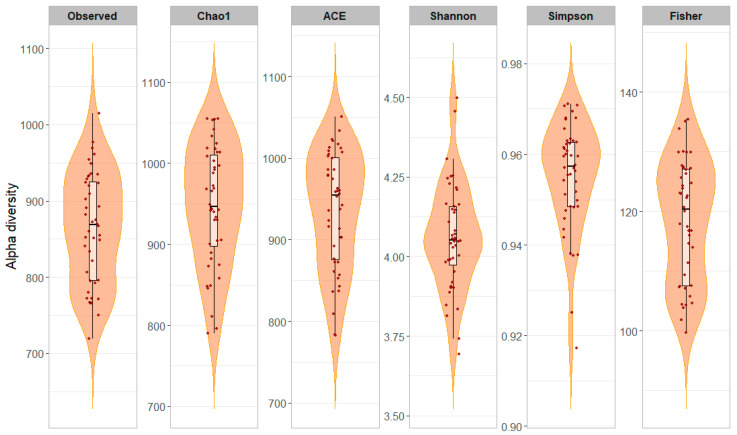
Alpha diversity of the milk microbiota across 44 individual samples. Each panel represents one of six diversity indices (Observed, Chao1, ACE, Shannon, Simpson, and Fisher). Violin plots show the distribution of values for each index, boxplots indicate the median and interquartile range, and jittered points represent individual samples. The figure highlights inter-individual variability in microbial diversity across samples.

**Table 1 animals-15-03149-t001:** Number of pair reads after each filtering step, for 44 individual and three bulk milk samples of Girgentana goat.

Step	Individual Milk	Bulk Milk
Raw data	15,279,670	988,744
Cutadapt (filtering for primers)	14,934,289	965,742
Sickle (trimmed for Phred > 25)	14,605,890	946,415
Joining reads (default parameters)	9,545,050	628,637
Filtering for unknown bases	8,135,944	536,937

**Table 2 animals-15-03149-t002:** Mean values of alpha diversity indices calculated for individual and bulk milk samples.

Samples	Observed	Chao1	ACE	Shannon	Simpson	Fisher
Individual	864.39	948.10	936.29	4.06	0.95	118.45
Bulk	324	353.93	342.42	3.97	0.96	38.80

## Data Availability

The raw data supporting the conclusions of this article will be made available by the authors on request.

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
