# Peer review of "Girgentana’s Goat Milk Microbiota Investigated in an Organic Farm During Dry Season"

_animals, 2025, doi:10.3390/ani15213149_

Round 1

Reviewer 1 Report

Comments and Suggestions for Authors

The introduction should better contextualize the importance of the Girgentana breed within the global landscape of indigenous breeds and justify why microbiome analysis is strategic (animal health, traceability, milk quality). It is also necessary to include papers from 2023 e 2024 on the goat milk microbiome and the impact of production systems. The methodology should detail the herd size and animal selection criteria (age, lactation stage, health), specify the version and parameters of the Phyloseq and R software, describe negative and positive controls for DNA extraction and amplification (even if already referred to as "blank controls"), and indicate the reference database used in the RDP classifier (e.g., SILVA, Greengenes, or RDP release version).
The results should include statistical significance in diversity comparisons. Add a visual summary (stacked bar chart) with the main families or genera to complement the pie charts. The discussion is consistent, but lacks direct comparisons with other goat or sheep breeds.

Further explore the functional significance of the microorganisms found, such as potential probiotics (Lactobacillus, Bifidobacterium) and technological implications.

Furthermore, it is important to address limitations such as:

-Seasonality (sampling only in the dry season).

-Lack of temporal replication.

-Potential environmental sources of contamination.

-Suggest future perspectives, such as longitudinal studies or functional metagenomics.

For the conclusion, I suggest rewording it to be more analytical and less descriptive. Suggested example:

“This first characterization of the Girgentana goat milk microbiota revealed a diverse community dominated by lactic acid bacteria, suggesting adaptation to local semi-extensive conditions. Future integrating studies functional metagenomics and multi-seasonal sampling will be key to understanding how these microbial ecosystems contribute to product quality and breed resilience.”

Author Response

Comment 1: The introduction should better contextualize the importance of the Girgentana breed within the global landscape of indigenous breeds and justify why microbiome analysis is strategic (animal health, traceability, milk quality). It is also necessary to include papers from 2023 e 2024 on the goat milk microbiome and the impact of production systems.

Response 1: We thank the reviewer for this valuable suggestion. The Introduction section has been substantially revised to provide a broader context for the Girgentana goat within the global framework of indigenous breeds and to emphasize the strategic importance of microbiome research in relation to animal health, milk quality, and traceability. Recent studies on milk microbiota (> 2023) have been implemented.

Comment 2: The methodology should detail the herd size and animal selection criteria (age, lactation stage, health), specify the version and parameters of the Phyloseq and R software, describe negative and positive controls for DNA extraction and amplification (even if already referred to as "blank controls"), and indicate the reference database used in the RDP classifier (e.g., SILVA, Greengenes, or RDP release version).

Response 2: We thank the reviewer for these comments. The Materials and Methods section has been revised to include detailed information on herd size, animal selection criteria, software versions and parameters, quality controls, and the taxonomic reference database used for classification. In particular, the herd size and selection criteria (age, lactation stage, health status) have been specified (lines 108-113), the versions and parameters of R and Phyloseq have been reported (lines 188-189), controls used for DNA amplification (lines 146-148), and the RDP classifier version and reference database (lines 177-179).

Comment 3: The results should include statistical significance in diversity comparisons.

Response 3: We thank the reviewer for this observation. We fully agree that testing for statistical significance in diversity comparisons would strengthen the interpretation of the data. However, due to the limited number of bulk milk samples (n = 3), formal statistical testing would not provide reliable or meaningful results.

Comment 4: Add a visual summary (stacked bar chart) with the main families or genera to complement the pie charts.

Response 4: Thanks for the suggestion. We included a stacked barplot of Genera (Figure 1C) as a complementary information of Figure 1.

Comment 5: The discussion is consistent, but lacks direct comparisons with other goat or sheep breeds. Further explore the functional significance of the microorganisms found, such as potential probiotics (Lactobacillus, Bifidobacterium) and technological implications.

Response 5: Discussion section has been largely implemented with comparisons with other breeds, as well as technological implications of some microorganism species. We thank the reviewer for the suggestions.

Comment 6: Furthermore, it is important to address limitations such as:

- Seasonality (sampling only in the dry season).

- Lack of temporal replication.

- Potential environmental sources of contamination.

- Suggest future perspectives, such as longitudinal studies or functional metagenomics.

Response 6: We implemented some considerations in the final part of the Discussion section. Thanks.

Comment 7: For the conclusion, I suggest rewording it to be more analytical and less descriptive. Suggested example: “This first characterization of the Girgentana goat milk microbiota revealed a diverse community dominated by lactic acid bacteria, suggesting adaptation to local semi-extensive conditions. Future integrating studies functional metagenomics and multi-seasonal sampling will be key to understanding how these microbial ecosystems contribute to product quality and breed resilience.”

Response 7: Thanks. We re-wrote the Conclusion section according to the reviewer’s comment.

Reviewer 2 Report

Comments and Suggestions for Authors

L22 Suggestion, remove “offering insights into artisanal dairy systems” as this study cannot be representative for other dairy systems and “artisanal” rarely is used to describe the production form.

L65 Although I agree that semi-extensive farming system have scarce information on environmental factors the climate condition is not such factor as the local climate can be monitored.

L69. remove; ”sustainable farm management” – although expressed carefully I find it questionable how knowledge on milk microbiota can affect farm management sustainability.

L83-88. These are results and should be presented under the result section

L92-104. This section would benefit of higher clarity and more detail. 53 goats were sampled, how many goats were there in the herd? Were both teats sampled? Was all the sampling performed during one milking?

Clinical and sub-clinical mastitis are factors that in other studies have been shown to affect the milk microbiota, but no information on the health statues of the sampled goats are given.

Goats were kept in a semi-extensive management system yet their diet primary consisted of hay and whole grains. Were the goats allowed to graze? Was there an additional feed intake?

For clarity an explanation of the filtering step would be recommended.

A statement on ethical approval to use animals in research is missing.

L108. The description and DNA extraction and library preparation is at large detailed enough for replication. However, the volume of milk used to extract DNA is missing, likewise where the NovaSeq sequencing were performed.

Why was not a positive control ( a mock community) included?

L268 and L277, the sentences are repetitive.

L299. “(n=3) mat” should be “may”

L330 a common core taxa, that exist in all samples, is not described in the manuscript and cannot as such be included in the conclusion.

Author Response

Comment 1: L22 Suggestion, remove “offering insights into artisanal dairy systems” as this study cannot be representative for other dairy systems and “artisanal” rarely is used to describe the production form.

Response 1: We thank the reviewer for the clarification. We re-wrote the sentence.

Comment 2: L65 Although I agree that semi-extensive farming system have scarce information on environmental factors the climate condition is not such factor as the local climate can be monitored.

Response 2: We thank the reviewer for this helpful comment. We added a detailed environmental report at lines 114-119. These measurements provide environmental context.

Comment 3: L69. Remove” sustainable farm management” – although expressed carefully I find it questionable how knowledge on milk microbiota can affect farm management sustainability.

Response 3: Thanks for the suggestion. The entire section has been revised according to other reviewer’s comments.

Comment 4: L83-88. These are results and should be presented under the result section

Response 4: We removed these lines from the Introduction section.

Comment 5: L92-104. This section would benefit of higher clarity and more detail. 53 goats were sampled, how many goats were there in the herd? Were both teats sampled? Was all the sampling performed during one milking? Clinical and sub-clinical mastitis are factors that in other studies have been shown to affect the milk microbiota, but no information on the health statues of the sampled goats are given. Goats were kept in a semi-extensive management system yet their diet primary consisted of hay and whole grains. Were the goats allowed to graze? Was there an additional feed intake? For clarity an explanation of the filtering step would be recommended. A statement on ethical approval to use animals in research is missing.

Response 5: We thank the reviewer for these detailed comments. We have thoroughly revised the section “2.1 Animals and Sample Collection” to provide a clearer and more detailed description of the farm, the sampled herd, and the sampling procedure. As regards for the statement on ethical approval, this study does not fall under Legislative Decree 26/2014 on the protection of animals used for scientific purposes as it does not involve procedures that cause suffering or stress to the animals, but milk samples derived from normal milking practices adopted on farms for non-experimental purposes were collected. Ethical Information section has been implemented in the main text.

Comment 6: L108. The description and DNA extraction and library preparation is at large detailed enough for replication. However, the volume of milk used to extract DNA is missing, likewise where the NovaSeq sequencing were performed.

Response 6: We added additional information.

Comment 7: Why was not a positive control (a mock community) included?

Response 7: A mock community was not included in this study because the primary aim was to characterize and compare the microbial composition of milk samples rather than to perform method validation or absolute quantification. The DNA extraction, amplification, and sequencing workflows were conducted following standardized protocols that have been extensively validated in previous milk microbiota studies (DOI: 10.1128/Spectrum.00374-21). Therefore, while the inclusion of a mock community could have provided an additional layer of technical validation, the robustness and reproducibility of the employed methods ensure the reliability of the obtained taxonomic profiles.

Comment 8: L268 and L277, the sentences are repetitive.

Response 8: Thanks for the comment. We remove the redundancy in sentences.

Comment 9: L299. “(n=3) mat” should be “may”

Response 9: Thanks for the correction.

Comment 10: L330 a common core taxa, that exist in all samples, is not described in the manuscript and cannot as such be included in the conclusion.

Response 10: Thanks. We re-wrote this section entirely.

Round 2

Reviewer 2 Report

Comments and Suggestions for Authors

The amendments made to the manuscript has improved the overall impression and made the article suitable for publication